Practitioner training and user experience of Seeking Safety for people with complex post-traumatic stress disorder and substance use disorder

Marino Luise V. 1 2 3
Ogbeiwi Osahon 2
Mott Melanie 4
Jordan Matthew 4
Smith Tracey 2
Khan Wajid 2
Webber Martin martin.webber@york.ac.uk 5
1 School of Health Sciences, University of Huddersfield , Huddersfield , West Yorkshire , United Kingdom
2 Research & Development, South West Yorkshire Partnership NHS Foundation Trust , Wakefield , West Yorkshire , United Kingdom
3 School of Medicine, University of Southampton , Southampton , Hampshire , United Kingdom
4 Core Team West, South West Yorkshire Partnership NHS Foundation Trust , Wakefield , West Yorkshire , United Kingdom
5 School for Business and Society, University of York , York , United Kingdom
Abdullah Jafri
Electronic publication date: 2025 Sep 22
Publication date: 2025
Volume: 13
Electronic Location ID: e20010
Received 2025 Jan 10; Accepted 2025 Aug 7
Copyright: ©2025 Marino et al.
Copyright year: 2025
Copyright holder: Marino et al.
License: This is an open access article distributed under the terms of the Creative Commons Attribution License, which permits unrestricted use, distribution, reproduction and adaptation in any medium and for any purpose provided that it is properly attributed. For attribution, the original author(s), title, publication source (PeerJ) and either DOI or URL of the article must be cited.
License URL: https://creativecommons.org/licenses/by/4.0/

Keywords: Complex post traumatic stress disorder, Substance use disorder, Mental health, Seeking safety, Group sessions, Health care professionals

Funding: The authors received no funding for this work.

==============================
Background

Post-traumatic stress disorder (PTSD) develops because of a profoundly traumatic experience such as combat situations, interpersonal violence, accidents, and natural disasters. Symptom manifestation may include recurring intrusive thoughts and memories, low mood, hypervigilance, disrupted sleep patterns, emotional dysregulation, and reduced attention span. Individuals affected by complex PTSD may withdraw from society or engage in harmful, risky and dangerous behaviours or develop substance use disorder (SUD). The purpose of this scoping review is to consider available evidence around the use of Seeking Safety as a treatment modality in individuals with complex PTSD and SUD. In particular it aims to identify the available evidence relating to Seeking Safety with regards to (i) gaps in knowledge around implementation; (ii) which healthcare professionals (HCPs) deliver Seeking Safety; (iii) knowledge and training required to deliver it; and (iv) the experience of individuals completing Seeking Safety treatment.

Methods

A scoping review methodology was used to identify qualitative, quantitative, and grey literature of Seeking Safety as a treatment modality in individuals with PTSD and SUD.

Results

A total of 451 studies were identified. Following deduplications, 431 records were screened for inclusion, the full-text of 24 articles were reviewed for eligibility and 18 were included in the review. Extracted data was synthesized and six overarching themes were identified: (i) Seeking Safety as a treatment; (ii) meeting the needs of a diverse patient population group; (iii) factors impacting success; (iv) empowerment of self and agency over life; (v) measuring treatment success; and (vi) knowledge gaps of Seeking Safety treatment.

Conclusion

This scoping review considers the gaps in knowledge around Seeking Safety, specifically relating to which HCPs are best suited to delivering it in clinical practice; the knowledge and training required to deliver it; and the experience of individuals undertaking Seeking Safety.

Introduction

Post-traumatic stress disorder (PTSD) develops because of a profoundly traumatic life experience such as combat situations, interpersonal violence, accidents, and natural disasters. Symptom manifestation may include recurring intrusive thoughts and memories, low mood, hypervigilance, disrupted sleep patterns, emotional dysregulation, and reduced attention span (Yehuda et al., 2015). In the United States of America (USA), a study of 9,463 adults aged 60 years and older, reported a prevalence rate of 46% of substance use disorder (SUD) among people with PTSD and the prevalence of alcohol abuse ranged from 9.8% to 61.3% (Pietrzak et al., 2012). Complex PTSD is a new diagnosis in ICD-11 (UK Trauma Council, 2024). In the UK, the one-month prevalence of complex PTSD is estimated to be between 1% and 8% in the general population, (Najavits, 2002) and the lifetime prevalence is 13% (Foa et al., 2016). In at-risk populations such as refugees seeking treatment, prevalence estimates for complex PTSD are higher ranging from 16% to 38% (Najavits, 2002). Symptoms of complex PTSD are likely to remit within 6 months of treatment, with the majority within 2 years (Schnurr et al., 2024). Many people with complex PTSD remain undiagnosed or untreated, inspiring other diagnoses which further delays the treatment of underlying trauma (Kaufman et al., 2016). In addition, co-morbidities including chronic pain, cardiometabolic disease and dementia are commonly associated with PTSD and SUD compounding the burden of disease and increasing mortality risk (Rosenbaum et al., 2015; Michopoulos et al., 2015).

Individuals with complex mental health problems often require multiple treatments; some treatment modalities may not entirely prevent future risk of developing a new disorder, leading to individuals repeatedly seeking support and treatment for unmet mental health needs (Lortye et al., 2021). People with complex PTSD present to National Health Services (NHS) mental health services with acute symptoms of trauma, which they may manage with alcohol, drugs, or other unsafe coping behaviours (Najavits, 2002). Outpatient psychological therapy is more cost effective (Ki et al., 2023), with exposure therapies (Coventry et al., 2020) and trauma-focused cognitive behavioural therapy (CBT) providing effective treatment packages for managing complex trauma (Coventry et al., 2020; Ennis, Sijercic & Monson, 2021). For some individuals, symptoms persist for many years as it is not possible to achieve psychological stability to access care (Rosellini et al., 2018). However, the treatment modalities require an individual to be psychologically and socially stable, so the process of therapy does not cause acute interpersonal destabilisation (Lortye et al., 2021). For some individuals with complex PTSD and SUD this may not be possible, and as a result some individuals end up in a cycle of hospital inpatient admissions which are distressing, costly and resource intensive (Kessler & Wang, 2008). As a result developing effective interventions supporting stabilisation for people with complex PTSD and SUD has been highlighted as a research priority by the National Institute for Health and Care Excellence (Megnin-Viggars et al., 2019), suggesting new approaches to treatments may be required to support sustained remission (Menzies et al., 2024).

Seeking Safety is a group treatment for people with complex PTSD and drug and/or alcohol use, and was developed in the United States (Najavits, 2002; Najavits, 2007; Najavits et al., 1998). Seeking Safety offers 25 topics based on five key domains: (i) safety as the overarching purpose (supporting individuals attain safety in their relationships, thoughts, behaviour, and emotions); (ii) integrated treatment (working on trauma and substance abuse concurrently); (iii) focus on ideals to oppose the loss of ideals in both PTSD and SUD; (iv) content areas, cognitive, behavioural, interpersonal, case management; (v) attention to clinician processes (healthcare professionals’ emotional responses, self-care). The 25 topics could be conducted in any order (although the effect of this has not been studied), with as few or many as the service permits (Najavits, 2007; Najavits et al., 1998).

Over the last twenty years Seeking Safety as a treatment modality has undergone efficacy and effectiveness studies including (i) individual sessions (Hien et al., 2015), group sessions (Boden et al., 2012; Crisanti et al., 2019), (ii) combined with adjunctive pharmacotherapy (Hien et al., 2015) or other supportive treatments (Murphy et al., 2019; Ragg, Soulliere & Turner, 2019), (iii) peer led delivery, case manager or clinicians (Crisanti et al., 2019; Desai et al., 2008), (iv) use in non-healthcare settings amongst a wide range of ethnically and socially diverse participants; such as military veterans (Boden et al., 2012; Desai et al., 2008), physical disability (Anderson & Najavits, 2014), homeless (Desai et al., 2008), men and women detained in the criminal justice system (Lynch et al., 2012; Zlotnick, Johnson & Najavits, 2009; Barrett et al., 2015), adolescent girls (Najavits, 2007), indigenous populations (Marsh et al., 2015), pregnant women (Shenai, Gopalan & Glance, 2019), and transgender women (Empson et al., 2017; Takahashi et al., 2022). A recent meta-analysis evaluated the effect of Seeking Safety on PTSD symptoms and SUD, exploring relationships with time and the dose–response of Seeking Safety versions e.g., abbreviated to full version (Sherman et al., 2023). The meta-analysis suggested (i) an abbreviated number session of 6–15 were as effective as the full version 25 session, (ii) the most impactful sessions appeared to be: introduction to safety; compassion; detaching from emotional pain (grounding); safety; when substances control you; red and green flags; asking for help; setting boundaries in relationships; self-nurturing; taking back your power for PTSD; commitment; recovery thinking; coping with triggers; honesty; integrating the split self; healing from anger and termination; creating meaning; and (iii) session duration were 60–90 min although there is considerable variance in study results. The majority of studies in a wide variety of settings, considered groups sessions offered single sex groups (Boden et al., 2012; Zlotnick, Johnson & Najavits, 2009; Schäfer et al., 2019; Hien et al., 2004; Hien et al., 2009; Cash Ghee, Bolling & Johnson, 2009), with mixed sex groups in only one study (Najavits et al., 2014). Studies used a variety of outcome scales including Addiction Severity Index (ASI) (Boden et al., 2012; Schäfer et al., 2019; Najavits et al., 2018) and Clinical Assessed PTSD Scale (CAPS) v.1 (Blake et al., 1995), although across studies there was a wide variation with regards to measures, timepoints for assessing PTSD and substance use disorder making it challenging to develop dose–response curves (Sherman et al., 2023).

Although, Seeking Safety is well studied within a variety of care settings in the USA there is a paucity of information regarding the efficacy as a treatment modality of Seeking Safety amongst individuals with a dual diagnosis of complex PTSD and SUD within the NHS in the United Kingdom (UK), and how this treatment modality could be implemented at scale. This review aims to identify gaps in knowledge regarding Seeking Safety, specifically: (i) implementation, (ii) the healthcare professionals responsible for delivering it, (iii) the training required, and (iv) the experiences of individuals who complete the treatment.

Methods

Design

A scoping review methodology was used to identify qualitative, quantitative, and grey literature of Seeking Safety as a treatment modality in individuals with complex PTSD and SUD to help identify gaps in the literature and future research priorities (Arksey & O’Malley, 2005). A scoping study design was selected as it allows for the identification and synthesis of diverse evidence sources while mapping key concepts related to Seeking Safety treatment (Pollock et al., 2024). Multiple searches, with a content analysis and charting of data extracted completed. The data of interest included country of study, study methodology, population of interest, the role of HCPs, group composition, experience of participants, training and summary of key findings. The scoping review methodology provided an opportunity to focus on Seeking Safety as a treatment modality for individuals with PTSD and SUD and help identify gaps in the literature and future research priorities (Pollock et al., 2024; Peters et al., 2015). The Preferred Reporting Items for Systematic reviews and Meta-Analyses extension for Scoping Reviews (PRISMA-ScR) was used to structure the review (Tricco et al., 2018).

Research questions

• How has Seeking Safety been implemented within clinical practice within a National Health Service?

• Which healthcare professionals most commonly deliver Seeking Safety sessions?

• What key knowledge and training is required for healthcare professionals providing Seeking Safety as a treatment modality?

Current knowledge and practice

Using the PRISMA checklist (Arksey & O’Malley, 2005; Tricco et al., 2018) an a priori scoping review protocol was developed, and included (1) the research question, (2) eligibility criteria of the studies be to included, (3) information sources to be searched, (4) description of a full electronic search strategy, (5) data charting process with data items included, (6) critical appraisal and synthesis of the data in order to answer the question posed.

Data sources

A search across seven databases (Medline, Cochrane, EMBASE, CINAHL, SCOPUS, Web of Science, PubMed) was conducted, covering studies published from April 2013 to June 2025. This timeframe ensures the inclusion of contemporary research while maintaining a manageable scope. Forward and backward citation searching was completed on studies exploring the use of Seeking Safety as a treatment modality in individuals with complex PTSD and SUD.

Search strategy

A search strategy was devised with the assistance of an information specialist, using key words from the grey literature and modified for additional electronic data bases (Supplementary File: Table S1). The initial charting form was revised in Microsoft Excel (Redmond, Washington, USA) and tested independently by two team members (LVM, OO) using at least three sources (Ewald et al., 2022). LVM and OO independently reviewed titles, abstracts, and full texts to reduce bias, and conflicts were resolved by a third reviewer, MW. Following an iterative process and team agreement, only pertinent information was extracted, including details on the evidence source (author(s), year of publication, country), study information (design, population, study location, seeking safety, aims, results and research gaps. This review did not critically appraise evidence sources, as scoping reviews map and explore available evidence, rather than evaluate interventions or change clinical practice (Tricco et al., 2018; Munn et al., 2018). In addition, sources were searched for any references which may fulfil the inclusion criteria.

Study selection

Duplicates were deleted, titles and abstracts were screened, and full text articles were reviewed for eligibility. The inclusion criteria were: studies of adults which used a qualitative or quantitative design relating to the delivery of Seeking Safety treatment for individuals with complex PTSD and SUD co-morbidity; studies which relate to either professionals delivering intervention and/or experience of those receiving the intervention; studies in English.

The exclusion criteria were: children <18 years of age; abstracts; letters to the editor; systematic and meta-analysis reviews; studies relating to Seeking Safety which do not address our research questions or those that did not relate to the management of complex PTSD and SUD co-morbidity (i.e., gambling, sexual addiction, violent behaviour, etc.); publications not in the time frame; publications not in English language.

Data extraction

Data extraction was completed using a data extraction template (Microsoft 2010, Redmond, WA, USA) which captured the study design, results (in relation to our research questions), and conclusions.

Collating, summarizing and reporting the results

Data synthesis was completed using an established content analysis approach (Hsieh & Shannon, 2005), which was chosen as a technique for reporting common themes within data (Vaismoradi, Turunen & Bondas, 2013). Descriptive aspects of the study, methodology, outcomes, and relevant findings were coded. A conceptual framework was developed by creating initial codes, sub-categories, and then overarching themes which formed the resultant synthesis of findings into a content framework. This was done in a three-stage process with—stage 1 an initial sample of 113 codes was open coded into broad comment categories to develop an initial framework by two researchers (LVM, OO); stage 2—the best fit framework guided the categorization of all comments from the data (LVM), with further refinements (LVM, OO); and stage 3—overarching themes were identified (LVM, OO). The number of codes were counted to identify the weight of the themes from which sub-themes and over-arching themes were developed. Where there was conflict a third researcher (MW) was consulted.

Results

Selection and characteristics of included articles

A total of 456 records were identified, including information from the grey literature. Following the removal of duplicate records, abstracts and titles of 436 records were screened for inclusion. The full text of 24 articles were reviewed for eligibility, of which 18 were included in the analysis (Fig. 1).

Figure 1 PRISMA flow diagram.

Study characteristics

Of the 18 studies reviewed, 14 were conducted in the USA (Boden et al., 2012; Anderson & Najavits, 2014; Empson et al., 2017; Takahashi et al., 2022; Najavits et al., 2014; Najavits et al., 2018; Airdrie, Lievesley & Griffith, 2021; Bauer et al., 2022a; Morgan-Lopez et al., 2013; Morgan-Lopez et al., 2014a; Najavits et al., 2024; Salvador et al., 2020; Zaccari et al., 2017; Holman et al., 2020), two in Germany (n = 2) (Schäfer et al., 2019; Kaiser et al., 2015), one in the United Kingdom (UK) (Airdrie, Lievesley & Griffith, 2021), and one from Australia (Barrett et al., 2015). Of these two were qualitative studies (Airdrie, Lievesley & Griffith, 2021) exploring the experience of women with PTSD and SUD attending a Seeking Safety group, and as part of peer led Seeking Safety groups (Najavits et al., 2014). Three were secondary analysis of data arising from National Institute on Drug Abuse Clinical Trials network, examining the dose (Anderson & Najavits, 2014) and synergy with post-treatment twelve step affiliation on direct and indirect effects of PTSD and SUD in women (Morgan-Lopez et al., 2013; Morgan-Lopez et al., 2014a). Others included the use of Seeking Safety among women during the perinatal/ post-partum period (Salvador et al., 2020), transgender women (Takahashi et al., 2022), women (Schäfer et al., 2019; Bauer et al., 2022a; Kaiser et al., 2015) with HIV (Empson et al., 2017), mixed-sex groups (Zaccari et al., 2017; Najavits et al., 2016), military veterans (Najavits et al., 2016; Tyler Boden et al., 2014), individuals within the justice system (Boden et al., 2012; Barrett et al., 2015; Najavits et al., 2018; Holman et al., 2020) and as a mobile application (Najavits et al., 2024).

Content analysis: conceptual framework and overarching themes

A content analysis identified 113 codes, 11 sub-themes and five overarching themes, which were identified (Supplementary file: Table S1) as:

1. Seeking safety as a treatment; (i) contents of sessions, (ii) number of sessions, (iii) standardisation,

2. Meeting the needs of a diverse patient population group; (i) ideal group composition (i.e., single-sex versus mixed-sex groups),

3. Factors impacting success; (i) interplay between PTSD and SUD, (ii) treatment fidelity—sessions and attendance,

4. Empowerment of self and agency over life involves strengthening foundation of the self; (i) agency over self, (ii) group social support and validation,

5. Knowledge gaps of Seeking Safety treatment; (i) what are the mechanisms of action and what leads to sustained remission, (ii) what training is required to successfully deliver the treatment, (iii) what, how, when and who would benefit from Seeking Safety treatment?

These were used to develop a conceptual framework of the interdependencies of Seeking Safety (Fig. 2).

Figure 2 Conceptual framework of the interdependencies of Seeking Safety treatment.

Demographics of study participants

A total of 1,553 adults with complex PTSD and SUD comorbidity were included in the 18 studies (Boden et al., 2012; Anderson & Najavits, 2014; Barrett et al., 2015; Empson et al., 2017; Takahashi et al., 2022; Schäfer et al., 2019; Najavits et al., 2014; Najavits et al., 2018; Airdrie, Lievesley & Griffith, 2021; Bauer et al., 2022a; Morgan-Lopez et al., 2013; Morgan-Lopez et al., 2014a; Najavits et al., 2024; Salvador et al., 2020; Zaccari et al., 2017; Holman et al., 2020; Kaiser et al., 2015) of which 93.6% (n = 1455/1553) were female. The mean age of participants was 39.6 ± 9.2 years of which 80.6 ± 31.2% were females. The average number of sessions offered were 18.4 ± 6.7 (range: 8, 25), over 12.1 ± 7.2 weeks (range: 3, 25), 1.3 ± 0.5 times per week (range: 1, 2) for 77.1 ± 16 min (range: 60, 90). Only one study offered individual sessions (Najavits et al., 2018) with the remainder as single sex groups (Boden et al., 2012; Anderson & Najavits, 2014; Barrett et al., 2015; Empson et al., 2017; Takahashi et al., 2022; Schäfer et al., 2019; Najavits et al., 2014; Airdrie, Lievesley & Griffith, 2021; Bauer et al., 2022a; Morgan-Lopez et al., 2013; Morgan-Lopez et al., 2014a; Salvador et al., 2020; Holman et al., 2020; Kaiser et al., 2015), except for two which was offered as a mixed-sex group (Zaccari et al., 2017; Najavits et al., 2016) (Table S2).

Seeking safety as a treatment

(i) Contents of sessions

Where fewer than 25 sessions were offered the choice of modules varied, and did not have to completed in order, with the most common being: PTSD: taking back your power, Detaching from emotional pain: Grounding, When Substances control you, Taking good care of yourself, Compassion, Honesty, Integrating the split self, Commitment, Setting boundaries in relationships, Respecting your time, Healthy relationships, Healing from Anger (Empson et al., 2017; Schäfer et al., 2019; Najavits et al., 2018; Najavits et al., 2024). Some participants reported grounding was the most useful module (Empson et al., 2017), but found Honesty the most difficult (Empson et al., 2017; Takahashi et al., 2022).

(ii) Number of sessions

The average number of sessions offered were 18.4 ± 6.7 (range: 8, 25), over 12.1 ± 7.2 weeks (range: 3, 25), 1.3 ± 0.5 times per week (range: 1, 2) for 77.1 ± 16 min (range: 60, 90). Participants felt longer sessions in the beginning and more sessions 16 vs. 8 would have been useful. There was not enough time to discuss matters in detail and follow-up sessions would have been of benefit. Participants would have liked to have more opportunity to discuss their trauma, and more discussion on anxiety (Barrett et al., 2015).

(iii) Standardisation

All sessions offered followed the Seeking Safety treatment manual as developed by Najavits et al., and this was a condition of permission given to use Seeking Safety as a treatment (Najavits, 2007; Najavits et al., 1998). For the studies conducted in Germany (Schäfer et al., 2019; Kaiser et al., 2015), the manual was translated into German. However, six studies did not report any healthcare professional training (Anderson & Najavits, 2014; Empson et al., 2017; Takahashi et al., 2022; Morgan-Lopez et al., 2013; Morgan-Lopez et al., 2014a; Najavits et al., 2024).

Meeting the needs of a diverse patient population group

(i) Ideal group composition

Only one study offered individual sessions (Najavits et al., 2018) with the remainder as single sex groups (Boden et al., 2012; Anderson & Najavits, 2014; Barrett et al., 2015; Empson et al., 2017; Takahashi et al., 2022; Schäfer et al., 2019; Najavits et al., 2014; Airdrie, Lievesley & Griffith, 2021; Bauer et al., 2022a; Morgan-Lopez et al., 2013; Morgan-Lopez et al., 2014a; Salvador et al., 2020; Holman et al., 2020; Kaiser et al., 2015), except for two which was offered as a mixed-sex group (Zaccari et al., 2017; Najavits et al., 2016). No ideal composition of groups or size of the group were identified. Seeking Safety treatment has been delivered to individuals from a wider range of backgrounds in the USA. These studies have included individuals from different ethnicities, cultures and sex (Boden et al., 2012; Anderson & Najavits, 2014; Empson et al., 2017; Takahashi et al., 2022; Najavits et al., 2014; Najavits et al., 2018; Airdrie, Lievesley & Griffith, 2021; Bauer et al., 2022a; Morgan-Lopez et al., 2013; Morgan-Lopez et al., 2014a; Najavits et al., 2024; Salvador et al., 2020; Zaccari et al., 2017; Holman et al., 2020). However, there is limited evidence of treatment use in Germany (Schäfer et al., 2019; Kaiser et al., 2015), UK (Airdrie, Lievesley & Griffith, 2021), and Australia (Barrett et al., 2015).

Factors impacting success

(i) Interplay between PTSD and SUD

Seeking Safety as a treatment may have the potential to reduce PTSD and depression symptom severity, especially with an increasing number of sessions. Half of the participants did not complete a six-month follow-up interview. Some studies suggest that individuals lost to follow-up may have similar or better health outcomes than those who remain in the study. There is also the chance they may not have benefited from Seeking Safety as a treatment modality (Salvador et al., 2020). As military hospitals have short lengths of stay for complex PTSD and/or substance use disorder, a model that is feasible to deliver within this time frame is important. There were positive results for domains of substance use, PTSD symptoms, functioning psychopathology, and coping. The notable aspects of Seeking Safety was conducted in a short time frame averaging 11 sessions (Najavits et al., 2016). Participants enrolled within a peer-led group delivering Seeking Safety liked the goal-orientated, positive coping, behaviour modification of the treatment programme. Participants of these groups disliked the long check-in and short check-out, and did not like it when counsellors chose to sit in (Najavits et al., 2014).

(ii) Treatment fidelity—sessions and attendance

Kaiser et al. (2015) reported 38% (n = 20/33) of the participants did not meet the minimum doses, requiring further research into the reasons for poor treatment compliance. This is contrasted by women with high attendance >90% of Seeking Safety groups had joint in-treatment impact on PTSD and substance use disorder, with a sustained post-treatment change in PTSD and substance use disorder. Women who attended almost of the available Seeking Safety sessions showed reduction in alcohol or cocaine use that were transmitted through reduction in PTSD symptoms at 12 months (Morgan-Lopez et al., 2013). However, only 30–31% of the group were completers attended 90% of treatment, 27% were titrators achieving 50–80% attendance of classes through to the 7th session, and 42–43% were drop-outs and did not attend treatment beyond the 4th session. The results of this study suggest the largest decline in alcohol use rates in women in the Seeking Safety group were those who had the greatest attendance of a sobriety support group e.g. 12-step programme (Morgan-Lopez et al., 2014a). Symptoms of PTSD decreased over time in both men and women. Participants (both men and women) who attended up to 12 modules over 12 weeks showed non-significant greater reductions in PTSD symptoms compared to those who attended nine or less (Zaccari et al., 2017).

Empowerment of self and agency over life involves strengthening foundations of the self

(i) Agency over self

For women attending Seeking Safety groups, sessions provided the opportunity for: (i) strengthening foundations of the self, (ii) understanding of self and role of substances, (iii) alternative perspectives, (iv) empowerment, agency and activity, (v) evocation and management of emotions, (vi) safety and validation provided relationally, and a (vii) facilitator as a container (Airdrie, Lievesley & Griffith, 2021). For men, all components of treatments were useful, but some felt sessions were not long enough which did not leave enough time to discuss matters in detail. Some men reported they would have liked to discuss the traumatic experience to help them vent (Barrett et al., 2015).

(ii) Group social support and validation

Participants valued the teaching of concrete, practical and simple skills highlighting grounding as especially helpful. However, an important component appears to be group dynamics and the impact of the facilitator or if a second facilitator was present, or new members joined the group (Airdrie, Lievesley & Griffith, 2021). Themes arising from qualitative interview describe a number of group experiences, with several overarching themes; (i) group connection as a source of support and validation, (ii) challenges to relational safety, (iii) readiness and commitment, (iv) preparedness and ability to commit, (v) concrete, practical and simple, (vi) focusing on the present, (vii) consolidation—rather than new content, (viii) skill of the facilitator, and that (ix) Seeking Safety is not an island (Airdrie, Lievesley & Griffith, 2021). Thoughts on attending a peer led Seeking Safety offered the following insights. Participants were positive about it being peer led, reinforcing one addict helping another addict. Empowering for peers (Najavits et al., 2024). Wolff et al. (2015) reported the aspects participants liked the least was the disbanding of their group, with many reporting the end of it was traumatic.

Knowledge gaps of Seeking Safety treatment

(i) What is the mechanism of action and what leads to sustained remission?

Further research is required to explore the use of mixed gender vs. single gender groups, as well as tracking symptom changes after each group to provide more precise information about the minimum effective dose (Zaccari et al., 2017). Several studies paid incentives from $20–$50 per assessment (Takahashi et al., 2022; Schäfer et al., 2019; Najavits et al., 2018) and $3 per biological samples (Najavits et al., 2018). End of treatment affiliation with sobriety programmes suggest a possible mechanism of action to sustained remission. The largest decline in alcohol use rates in women in the Seeking Safety group were those who had the greatest attendance of the 12-step programme compared to those who did not attend the programme. This suggests there may be some synergy between these two programmes and may explain why some women in other studies do poorly. These post-treatment programmes are very accessible, offering social support. However, they are offered in mixed groups with women often only making up 35% of groups (Morgan-Lopez et al., 2013; Morgan-Lopez et al., 2014b). The extent to which mixed groups impact on women’s entry into this type of post-treatment support is unknown but is likely to be an important question to explore as part of future work. There are limitations to this work with regards to the lack of intention to treat study design.

(ii) What training is required to deliver the treatment?

In six of the studies no training was noted (Anderson & Najavits, 2014; Empson et al., 2017; Takahashi et al., 2022; Morgan-Lopez et al., 2013; Morgan-Lopez et al., 2014b). In other studies Seeking Safety was delivered by a variety of healthcare professionals including substance abuse counsellors (Schäfer et al., 2019; Holman et al., 2020), psychiatrists, social workers, occupational therapists or nurses (Schäfer et al., 2019), clinical psychologist (Barrett et al., 2015), psychotherapist (Airdrie, Lievesley & Griffith, 2021) with a Master degree (Bauer et al., 2022b), PhD (Boden et al., 2012; Kaiser et al., 2015), a certified Seeking Safety trainer (Schäfer et al., 2019; Najavits et al., 2018; Najavits et al., 2016). Fidelity to the treatment was also measured in several studies with 5–10% of sessions reviewed by a certified Seeking Safety practitioner (Schäfer et al., 2019; Najavits et al., 2018; Salvador et al., 2020; Zaccari et al., 2017; Najavits et al., 2016).

However, the ideal staff mix required, number of times per week it is offered or whether it is delivered by one or two healthcare professionals per session has not been determined. Holman et al. (2020) report as Seeking Safety is a manualised treatment protocol, it is inexpensive to implement and can be led by trained paraprofessionals.

(iii) What, how, when and who would benefit from Seeking Safety treatment?

Higher education level predicted greater alliance, controlling for age and income. In addition, a greater number of substance treatment attempts predicted higher earlier alliance. Education level significantly predicted treatment feedback when controlling for alliance. Integration of pre-treatment alliance building strategies with women from ethnic minority groups who have lower levels of education, through personalized discussions and goal setting with therapists to develop a strong partnership, may support better outcomes (Bauer et al., 2022b). In contrast where women were found to have lower levels of literacy, reducing the number of handouts and promoting group participation through role play improved the quality of sessions (Empson et al., 2017). For some studies several participants (15%) dropped out before treatment started or did not meet the minimum doses (38%), suggesting requiring further research into the reasons for poor treatment compliance (Kaiser et al., 2015).

Discussion

The results of this scoping review suggest Seeking Safety as a treatment modality has a role to play in supporting individuals with complex PTSD and SUD. From the available evidence five overarching areas where there are gaps in our knowledge were identified, including (i) Seeking Safety as a treatment; (ii) Meeting the needs of a diverse patient population group; (iii) Factors impacting success; (iv) Empowerment of self and agency over life involves strengthening foundation of the self, (v) Measuring treatment success; and (iv) Knowledge gaps of Seeking Safety treatment. Only one study offered individual sessions (Najavits et al., 2018) with the remainder as single sex groups (Boden et al., 2012; Anderson & Najavits, 2014; Barrett et al., 2015; Empson et al., 2017; Takahashi et al., 2022; Schäfer et al., 2019; Najavits et al., 2014; Airdrie, Lievesley & Griffith, 2021; Bauer et al., 2022a; Morgan-Lopez et al., 2013; Morgan-Lopez et al., 2014a; Salvador et al., 2020; Holman et al., 2020; Kaiser et al., 2015), except for two which was offered as a mixed-sex group (Zaccari et al., 2017; Najavits et al., 2016). Seeking Safety treatment has been delivered to individuals from a wider range of backgrounds, ethnicity, cultures, genders within the USA (Boden et al., 2012; Anderson & Najavits, 2014; Empson et al., 2017; Takahashi et al., 2022; Najavits et al., 2014; Najavits et al., 2018; Airdrie, Lievesley & Griffith, 2021; Bauer et al., 2022a; Morgan-Lopez et al., 2013; Morgan-Lopez et al., 2014a; Najavits et al., 2024; Salvador et al., 2020; Zaccari et al., 2017; Holman et al., 2020). However, there is limited evidence of treatment use in Germany (Schäfer et al., 2019; Kaiser et al., 2015), UK (Airdrie, Lievesley & Griffith, 2021), and Australia (Barrett et al., 2015). In addition, most studies considered the use of Seeking Safety treatment in women (93.6%), suggesting further work is required to explore the use of this as a treatment modality in mixed groups as well as men only groups. From the studies included in this scoping review, no ideal composition of groups or size of the group were identified, nor the duration or whether post-treatment programmes support ongoing remission. Of interest, important factors for better outcomes from Seeking Safety were higher level of education and more treatment attempts with being which were associated with higher earlier alliance (Bauer et al., 2022b), where session moderators were able to adapt reading materials or the way in which the sessions were delivered i.e., using greater group participation satisfaction with the sessions were greater (Empson et al., 2017). It is not clear whether these types of adaptations affect fidelity of the manualized approach to Seeking Safety, but considerations around participants learning styles would appear to be an important contributing factor to the success of the treatment.

Morgan-Lopez et al. (2013) considered the synergy between adjunctive treatment post-Seeking Safety treatment and sustained remission. The results of this study reported the largest decline in alcohol use rates in women in the Seeking Safety group were those who had the greatest attendance of the 12-step sobriety programme compared to those who did not attend the meetings. This suggests there may be some synergy between these two programmes and may explain why some women in other studies do poorly (Kaiser et al., 2015). Post-treatment programmes are very accessible, offering social support (Morgan-Lopez et al., 2013; Morgan-Lopez et al., 2014b). However, they are offered in mixed groups with women often only making up 35% of groups. The degree to which mixed groups of 12-step or recovery programmes impacts on women’s joining this type of post-treatment support is unknown and is likely to be an important question to explore as part of future work (Morgan-Lopez et al., 2014b).

For some studies there was poor adherence to groups, with over one third not meeting the minimum treatment dose (Kaiser et al., 2015) or disengagement prior to treatment starting (Schäfer et al., 2019). From these studies it is not clear the reasons for poor treatment compliance, although factors such low levels of literacy (Takahashi et al., 2022), number of years of education (Bauer et al., 2022b) and poverty (Takahashi et al., 2022; Schäfer et al., 2019; Najavits et al., 2018) may impact treatment compliance. Identifying which participants will benefit most from this type of group work will be important for the sustainability of clinical services in the future (Schäfer et al., 2019; Najavits et al., 2018), especially as the need for mental health services within the National Health Service (NHS), UK is reported to be at an all-time high. NHS digital figures for 2020/21 reported the 634,649 individuals referred themselves for talking therapy for support with depression, stress and anxiety, completing on average 7.5 sessions (NHS England, 2024).

The studies included within this scoping review reported an average number of sessions offered were ranged from a partial dose of 8 sessions to the full dose of 25 sessions, delivered over 3 to 25 weeks once to twice per week (Boden et al., 2012; Anderson & Najavits, 2014; Barrett et al., 2015; Empson et al., 2017; Takahashi et al., 2022; Schäfer et al., 2019; Najavits et al., 2014; Najavits et al., 2018; Airdrie, Lievesley & Griffith, 2021; Bauer et al., 2022a; Morgan-Lopez et al., 2013; Morgan-Lopez et al., 2014a; Najavits et al., 2024; Salvador et al., 2020; Zaccari et al., 2017; Holman et al., 2020; Kaiser et al., 2015). The minimum dose required to overcome symptoms of complex PTSD and aid recovery is unknown, and whether any mode of psychological treatment would be as effective as another (Menzies et al., 2024). For example, Creating Change is a past-focused approach versus Seeking Safety a present-focused approach, has similar impact on reducing symptoms of complex PTSD and substance use disorders. It may be that future services are able to offer a more personalised approach to treatment based on individual preferences, as well as delivering topics identified as being essential to supporting sustained recovery. (Najavits et al., 2018). In their meta-analysis of Seeking Safety Sherman et al. (2023) report there are significant variations in the delivery and dose across the studies including topics covered. Although the full doses of Seeking Safety had better outcomes for complex PTSD and SUD at 3 months, there were no other statistically significant differences between the full and abbreviated versions of Seeking Safety. Findings from their meta-analysis show long term effect (more than 3 months) was observed with an abbreviated of Seeking Safety compared to the full dose (25 sessions). These findings have important implications for the implementation of Seeking Safety into routine clinical practice, especially within the NHS where there are high levels of unmet need for mental health services (McDaid et al., 2022).

As Seeking Safety does not process trauma or substance use disorder, it has been postulated that additional therapeutic skills by a certified or licensed professional may not be required (Holman et al., 2020). However, there may always be a risk members of the group are ‘triggered’ by some of the materials discussed with Seeking Safety sessions, leading to an escalation of mental health symptoms and associated behaviour such as hypervigilance, anger or disassociation. As a result there have been suggestions of a minimum level of professional training being to Masters or Doctoral levels (Holman et al., 2020), or at least be a certified Seeking Safety trainer (Schäfer et al., 2019; Najavits et al., 2018; Najavits et al., 2016). Six studies did not report any healthcare professional training (Anderson & Najavits, 2014; Empson et al., 2017; Takahashi et al., 2022; Morgan-Lopez et al., 2013; Morgan-Lopez et al., 2014a; Najavits et al., 2024) around Seeking Safety. In addition, the level of initial training, support and supervision for healthcare professionals has not been identified and within the studies this varied. Ongoing supervision for healthcare professionals involved in delivery of Seeking Safety sessions may be essential to reduce the risk of secondary post-traumatic stress disorder as a result of hearing survivors of traumatic events recount their lived experiences (Kizilhan, 2020; Fernandes, Rhodes & Buus, 2024). All of which may have implications for mental health services where health professionals have varying levels of competence and confidence and access to ongoing support or supervision (Rhodes et al., 2010). As such further research is required to better understand what essential knowledge, skills, competence and support is required of healthcare professionals to safely deliver Seeking Safety within this vulnerable population group.

Much has been made of the manualised approach of Seeking Safety and the ease to which it can be implemented, using counsellors as the clinicians delivering treatment (Takahashi et al., 2022; Najavits et al., 2016). Holman et al. (2020) report Seeking Safety as a manualised treatment protocol was very effective, inexpensive to implement and can be led by trained paraprofessionals. Evidence suggests other manualized approaches for PTSD such as exposure therapy and CBT are also cost-effective, (Tuerk et al., 2013; Dams, 2024). Despite this, the economic burden of PTSD and SUD on a society is considerable, with an estimated individual costs of $19,630 per person (2018) (Davis et al., 2022; von der Warth et al., 2020). As such further research is required to explore the cost-effectiveness (i.e., total costs and quality-adjusted life years) of Seeking Safety treatment across health and social care systems, on reducing health economic burden, health service utilization and improving longer term outcomes for individuals with complex PTSD and SUD.

Research limitations

There are number of limitations to this scoping review, particularly the imbalance between men and women, with most participants being women (93.6%) and the potential bias associated with the studies selected for this scoping review. As result the implication for clinical practice may be more suitable for women with complex PTSD and SUD. The efficacy in mixed-sex and male only groups requires further study. The quality of the evidence reported was not formally assessed in line with scoping review methodology but was noted to be varied with some small cohort sizes and significant heterogeneity within studies, notably the variability of measures used. and. This scoping review identified several gaps in the research including, variation with regards to the number of healthcare professionals’ delivery sessions, the characteristics of the clinicians’ providing sessions, whether single sex groups or individual sessions are best. There were also gaps in knowledge, skills, training, and education needs across the spectrum of healthcare professionals providing Seeking Safety, which requires further exploration. Finally, the optimal duration of treatment, number of sessions and topics is not well explored.

Conclusion

Seeking Safety as a treatment modality may be useful for individuals with complex PTSD and SUD, especially when adjunctive treatment is offered post-treatment. Although Seeking Safety is a promising treatment modality, there are many unanswered questions relating to the optimal duration of treatment, minimum dose and training requirements of healthcare professionals delivering sessions, particularly as part of routine clinical practice in a national health service. In addition, research is required to explore the use of Seeking Safety in mixed-gender groups and settings outside of the United States of America. A health economic analysis that includes both health and social care costs would provide a deeper, beneficial understanding of delivering this intensive support within routine clinical practice.

Supplemental Information

Supplemental Information 1 Preferred Reporting Items for Systematic reviews and Meta-Analyses extension for Scoping Reviews (PRISMA-ScR) Checklist

Supplemental Information 2 Development of codes, sub-categories and overarching themes

Supplemental Information 3 Characteristics of studies describing Seeking Safety treatment for individuals with complex PTSD and SUD

Additional Information and Declarations

Competing Interests

Author Contributions

Data Availability

The authors declare there are no competing interests.

Luise V. Marino conceived and designed the experiments, performed the experiments, analyzed the data, prepared figures and/or tables, authored or reviewed drafts of the article, and approved the final draft.

Osahon Ogbeiwi conceived and designed the experiments, performed the experiments, analyzed the data, authored or reviewed drafts of the article, and approved the final draft.

Melanie Mott analyzed the data, authored or reviewed drafts of the article, and approved the final draft.

Matthew Jordan analyzed the data, authored or reviewed drafts of the article, and approved the final draft.

Tracey Smith analyzed the data, authored or reviewed drafts of the article, and approved the final draft.

Wajid Khan analyzed the data, authored or reviewed drafts of the article, and approved the final draft.

Martin Webber conceived and designed the experiments, performed the experiments, analyzed the data, authored or reviewed drafts of the article, and approved the final draft.

The following information was supplied regarding data availability:

There is no raw data associated with this scoping review.

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
