# Peer review of "Practitioner training and user experience of Seeking Safety for people with complex post-traumatic stress disorder and substance use disorder"

_PeerJ, doi:10.7717/peerj.20010_

## Round 0.1 · original submission · Major Revisions

**Language Note:** The review process has identified that the English language must be improved. PeerJ can provide language editing services - please contact us at [email protected] for pricing (be sure to provide your manuscript number and title). Alternatively, you should make your own arrangements to improve the language quality and provide details in your response letter. – PeerJ Staff

·

Basic reporting

1. BASIC REPORTING
• Clarity and Language: The manuscript is written in professional English, but there are several instances where sentence structure could be improved for clarity and readability.
o Abstract: Page 1, Line 36-39: Sentence structure is fragmented and unclear: “It aims identify the available evidence relating to how Seeking Safety i) gaps in knowledge around implementation; ii) which healthcare professionals (HCPs) deliver Seeking Safety; and iii) knowledge and training required to deliver it; and v) the experience of individuals completing Seeking Safety treatment.”
o Suggestion: “This review aims to identify gaps in knowledge regarding Seeking Safety, specifically: (i) its implementation, (ii) the healthcare professionals responsible for delivering it, (iii) the training required, and (iv) the experiences of individuals who complete the treatment.”
o Rationale: Improved clarity and corrected numbering error (missing ‘iv’).
• Introduction & Background: The introduction provides a good overview of PTSD, SUD, and Seeking Safety. However, the research questions could be more clearly stated, and the justification for the scoping review could be more explicitly outlined.
o Example: “It aims to identify the available evidence relating to how Seeking Safety...” → The phrase is fragmented; consider restructuring.
• Page 2, Line 56-57, Overly complex sentence structure: “As a result, individuals affected by PTSD may withdraw from society or engage in harmful, risky and dangerous behaviours or develop substance use disorder (SUD) (2).”
• Suggestion: “Individuals affected by PTSD may withdraw from society, engage in risky behaviours, or develop substance use disorder (SUD) (2).”
• Rationale: More concise and readable.
• Some sentences are overly complex, making comprehension difficult. Line 69-71
o Example: “The revolving door nature of individuals with complex PTSD and SUD has a significant impact on health service provision, and often results in costly episodes requiring inpatient admissions.” → Consider simplifying for clarity. “The revolving door nature” is a metaphor that may not be immediately clear to all readers.
o Redundant Phrasing: “Has a significant impact on health service provision” could be simplified without losing meaning.
o Suggestion: The recurring cycle of individuals with complex PTSD and SUD frequently leads to costly inpatient admissions, straining healthcare resources
• Literature Referencing: The manuscript is well-referenced, drawing on relevant studies. However, there are some inconsistencies in citation formatting (e.g., missing spaces, incorrect use of parentheses, and inconsistent journal name formats).
o Example: "A study by Morgan-Lopez et al. (48, 58) considered the synergy between adjunctive treatment..." → Ensure consistency in citation formatting.
• Structure and Formatting: The structure generally follows scoping review guidelines, but could be better organized. Some sections contain redundant information, particularly in the methods and results sections.
o Example: Line 222, “Seeking Safety treatment has been delivered to individuals from a wider range of backgrounds, ethnicities, cultures, and genders within the USA.” → Consider breaking this into smaller, clearer sentences.
• Figures and Tables: Ensure that all figures and tables are correctly labeled and referenced within the text. No obvious issues with image manipulation were noted.

Experimental design

STUDY DESIGN
• Line 120-121: “A scoping study design was chosen because it offered a framework to identify and synthesize a broad range of evidence.”
• Suggestion: “A scoping study design was selected as it allows for the identification and synthesis of diverse evidence sources while mapping key concepts related to Seeking Safety treatment (41).”
• Rationale: Adds scholarly support and better explains methodology.
• Scoping Review Justification: The rationale for conducting a scoping review is justified, but the selection criteria for including studies need more clarity. Why was a 10-year timeframe chosen?
o Example: Line 138-142, “A ten-year time limit was set as April 2013 until April 2024, to ensure as much contemporaneous evidence was included.” → Provide a justification for this timeframe.
o Limited explanation of inclusion criteria: “A search was completed across seven electronic databases... A ten-year time limit was set from April 2013 until April 2024, to ensure as much contemporaneous evidence was included.”
o Suggested Revision: “A search across seven databases (MEDLINE, Cochrane, EMBASE, CINAHL, SCOPUS, Web of Science, PubMed) was conducted, covering studies published from April 2013 to April 2024. This timeframe ensures the inclusion of contemporary research while maintaining a manageable scope.”
o Rationale: Clearer justification for the selected time frame.
o
• Methodology Description:
o The methodology follows PRISMA-ScR guidelines but lacks details on how data extraction was conducted.
o The selection process could benefit from a flowchart to illustrate how records were filtered.
o Inclusion and exclusion criteria are appropriate, but could be more detailed regarding study types and populations.
o Example: Line 157-158, “Data extraction was completed using a data extraction template...” → Specify who performed the extraction and whether inter-rater reliability was assessed.

Validity of the findings

VALIDITY OF THE FINDINGS
Line 170-171
• Issue: Unclear description of included studies: “Most of the 18 studies were conducted in the USA (n=14) (20,25,31,32,38,39,46–53), followed by Germany (n=2) (34,54), United Kingdom (UK) (n=1) (46), and Australia (n=1) (28).”
• Suggestion: “Of the 18 studies reviewed, 14 were conducted in the USA, two in Germany, one in the UK, and one in Australia (20,25,31,32,38,39,46–53).”
• Rationale: More concise and clearer.
• Strength of Argument: The review summarizes key themes effectively but lacks critical analysis in some sections. More synthesis of findings, rather than just summarization, would strengthen the discussion.
o Example: "The five overarching themes identified include Seeking Safety as a treatment..." → Elaborate on why these themes were chosen over others.
• Themes Identified: The five overarching themes are well-structured, but some overlap exists between “Empowerment of self and agency over life” and “Meeting the needs of a diverse population.” Consider merging or distinguishing these more clearly.
o Example: "Empowerment of self and agency over life involves strengthening foundations of the self..." → Clarify if this overlaps with patient population needs.
Line 228-230: Unclear claim regarding attrition: “However, 50% of this group of participants did not complete a 6-month post-baseline interview. There is some evidence to suggest those lost to follow-up tend to be as healthy as or healthier than those completing follow-up interviews.”
• Suggestion: “Half of the participants did not complete a six-month follow-up interview. Some studies suggest that individuals lost to follow-up may have similar or better health outcomes than those who remain in the study.”
• Rationale: Clarifies the meaning and strengthens the claim.
• Discussion of Limitations: The limitations section acknowledges some gaps but does not discuss how the scoping review could impact clinical practice or research. It would benefit from addressing potential publication bias and language restrictions.
o Example: “There are limitations to this work, relating to Seeking Safety as a treatment modality for complex PTSD and SUD, with most participants being women.” → Discuss how this gender imbalance impacts findings.
• Future Research Directions: The conclusion mentions gaps in knowledge but does not provide specific recommendations for future research.
o Example: “Further research should explore the use of Seeking Safety in mixed-gender and non-US settings...” → Provide concrete recommendations.

Additional comments

GENERAL COMMENTS
• Strengths:
o The manuscript addresses an important topic with real-world implications for mental health treatment.
o The literature coverage is broad, capturing various settings and populations.
o The thematic synthesis provides a clear structure for findings.
• Weaknesses:
o Some sections (e.g., methods and discussion) are overly repetitive.
o Lacks in-depth critical analysis—more discussion is needed on the implications of findings.
o Some minor grammatical errors and awkward phrasing throughout.
o Needs more explicit discussion of limitations, including potential biases in study selection.

Reviewer 2 ·

Basic reporting

The study is well written, but the conclusion sentence could be written better. It does not cover the entire article. It would be better if the conclusions were rewritten.

When citing, I think there is no need to choose multiple sources.(20,25,28,31,32,34,38,46–49,51,53,54),

Experimental design

-

Validity of the findings

-

Additional comments

-

Annotated reviews are not available for download in order to protect the identity of reviewers who chose to remain anonymous.

Reviewer 3 ·

Basic reporting

-

Experimental design

-

Validity of the findings

-

Additional comments

The Results section in the abstract should be improved. The current text does not offer any result.

Reviewer 4 ·

Basic reporting

This manuscript is a scoping review of the Seeking Safety framework as a treatment for comorbid PTSD and SUD. The manuscript is poorly conceptualised and offers no new insight above and beyond the recent meta-analysis conducted by Sherman et al. (2023). In fact, conducting a scoping review after a meta-analysis is actually taking several steps backward. I have more detailed comments below, but I do not believe that this manuscript is conducted in an appropriate way, or worthwhile publication.

The abstract is poorly written in that it provides too much background information that belongs in the introduction, gives no information about the methods used to conduct the review, and is poorly written with several grammatical errors and incomplete sentences. In fact, sections are directly copied from the intro to the abstract.

Claims like this: “People with complex PTSD present to NHS mental health services with
acute symptoms of trauma, which they manage with alcohol, drugs, or other unsafe coping behaviours” need to be avoided. It is certainly not the case that everyone with PTSD manages their symptoms with alcohol or other drugs”.

Trauma-focused behavioural therapy is not the first choice of psychotherapy treatment for PTSD. Exposure therapies are much more commonly employed and require substantially shorter times.

Experimental design

The aims and research questions need to be carefully rephrased. At times, it sounds like just a replication study of the already published meta-analysis, whereas I believe the purpose is to specifically investigate the use of Seeking Safety in the NHS. This, however, does not align with the inclusion or exclusion criteria either. Moreover, some of the research questions are not actually phrased as questions.

Instead of listing the PRISMA criteria for a scoping review, it needs to be outlined HOW these criteria were addressed. I.e., how many reviewers completed the review, how were disagreements resolved, etc.

Validity of the findings

The review is over one year old. This needs to be updated.

Additional comments

Please write all sections of the manuscript in entire sentences and paragraphs.

---

## Round 0.2 · accepted · Accept

Thank you for your revised manuscript which is currently suitable to be accepted by Peer J. Congratulations.

·

Basic reporting

No Comment

Experimental design

No comment

Validity of the findings

No comment

Additional comments

The author has greatly improved on the manuscripts.

Reviewer 4 ·

Basic reporting

The authors have addressed my comments

Experimental design

The authors have addressed my comments

Validity of the findings

The authors have addressed my comments

Additional comments

The authors have addressed my comments